# Lentiviral Vectors as a Vaccine Platform against Infectious Diseases

**DOI:** 10.3390/pharmaceutics15030846

**Published:** 2023-03-05

**Authors:** Kirill Nemirov, Maryline Bourgine, François Anna, Yu Wei, Pierre Charneau, Laleh Majlessi

**Affiliations:** Pasteur-TheraVectys Joint Lab, Institut Pasteur, Université de Paris, 28 rue du Dr Roux, CEDEX 15, 75724 Paris, France

**Keywords:** lentiviral vaccine vectors, T-cell vaccines, tropism for dendritic cells, antigen presentation, major histocompatibility complex, inflammation, pre-existing anti-vector immunity

## Abstract

Lentiviral vectors are among the most effective viral vectors for vaccination. In clear contrast to the reference adenoviral vectors, lentiviral vectors have a high potential for transducing dendritic cells in vivo. Within these cells, which are the most efficient at activating naive T cells, lentiviral vectors induce endogenous expression of transgenic antigens that directly access antigen presentation pathways without the need for external antigen capture or cross-presentation. Lentiviral vectors induce strong, robust, and long-lasting humoral, CD8^+^ T-cell immunity and effective protection against several infectious diseases. There is no pre-existing immunity to lentiviral vectors in the human population and the very low pro-inflammatory properties of these vectors pave the way for their use in mucosal vaccination. In this review, we have mainly summarized the immunological aspects of lentiviral vectors, their recent optimization to induce CD4^+^ T cells, and our recent data on lentiviral vector-based vaccination in preclinical models, including prophylaxis against flaviviruses, SARS-CoV-2, and *Mycobacterium tuberculosis*.

## 1. Introduction

Gene transfer into the eukaryotic host cell is a powerful strategy for gene therapy and vaccination. Successful introduction of genetic material into cells can be achieved either by the use of transfection agents such as physical factors or non-viral particles, or by transduction using viral vectors. Among viral vectors, lentiviral vectors (LVs) show the greatest transduction capacity. LVs are enveloped, single-stranded (ss)RNA, non-replicating viral particles. Their strong transduction capacity is mainly related to a three-stranded DNA structure, known as a “DNA flap”, which is formed following retro-transcription of RNA into double stranded proviral DNA [1,2,3]. Transgenic genetic material of up to several kilobases can be inserted into an LV genome to be translocated into the host cell nucleus for transcription into mRNA and further translation into protein antigen(s). Integrative LVs may carry a risk of insertional mutagenesis and off target effects. However, for the use in vaccination and immuno-oncotherapy, this limitation is efficiently overcome by mutating the catalytic site of the LV integrase to generate non-integrative vectors. In this case, the DNA translocated into the cell nucleus remains in a circular episomal form, is nongenotoxic and does not interact with the host chromosome [4]. LVs are non-replicative, as all structural and functional genes from the HIV parental virus are excluded from the vector genome. LVs are not cytopathic and have no or very limited inflammatory properties, unlike the most widely used vaccine vectors, including adenoviral or modified vaccinia virus Ankara vectors [5,6].

In most cases, LVs are pseudotyped with the envelope glycoprotein of vesicular stomatitis virus (VSV-G), against which no significant pre-existing immunity has been described in human populations. LVs induce marked and durable humoral responses and especially high-quality memory-type CD8^+^ T-cell responses [4,7]. New generation LVs have been optimized to also induce robust CD4^+^ T-cell responses [8,9]. The marked capacity of these vectors to induce adaptive immune responses is linked to their notable in vivo tropism for dendritic cells (DCs). Inside these cells, which are the most effective at activating naïve T cells, LVs induce the efficient endogenous expression of transgenic antigens, which directly gain access to the antigen presentation pathways without the need of antigen capture from outside or cross presentation. This is in clear contrast to the reference adenoviral vectors that primarily target epithelial cells and therefore, require antigen capture by DCs and an indirect antigen presentation to trigger T-cell activation.

In this review, we first discuss the LV-based vaccine platform, the biological properties of LVs, their tropism for DCs and their biosafety. Then, we summarize our most recent results on LV-based vaccination in several preclinical models of several infectious diseases.

## 2. Lentiviral Vector-Based Vaccine Platform

The vast majority of LVs are derived from HIV-1. We will not detail here the genetic and structural characteristics of LVs, which we recently updated [4]. Briefly, all coding sequences from the parental HIV genome have been deleted in the vector system. LVs are therefore, strictly non replicative, allowing exclusive expression of the encoded transgene after a single cycle of cell transduction. Indeed, the genetic messages necessary for the production of LVs, which is mostly performed in the human embryonic kidney (HEK)293T cell line system, are distributed over at least three plasmids (Figure 1). (i) The packaging plasmid contains functional structural and *gag* genes and enzymatic *pol* genes, as well as the regulatory *tat* and *rev* genes. A third-generation LV has been described, with the genetic material distributed over four plasmids, instead of three. In these vectors, the regulatory *rev* gene is not in the packaging plasmid but rather in a dedicated independent plasmid. However, the production yield of these LVs is generally much lower [10], without providing any further safety. (ii) The envelope plasmid encodes the envelope glycoprotein, which is usually VSV-G. In these two plasmids, the genes are expressed under the transcriptional control of the cytomegalovirus immediate early promoter (P_CMV_). (iii) The transfer plasmid contains a Ψ; packaging signal, necessary for viral genome dimerization and packaging into the budding viral particles, the rev response element (RRE), involved in non-spliced RNA transport from the nucleus to the cytoplasm, and two central cis-active sequence specific to lentiviruses: the central polypurine tract (cPPT) and central termination sequence (CTS), required for the formation of the “DNA flap”, which is the source of the outperforming efficiency of nuclear translocation of the proviral DNA [11]. The DNA flap is formed following retro-transcription of RNA into double stranded proviral DNA through the use of cPPT and CTS [1,2,3]. The DNA flap triggers nuclear importation of the vector genome after the uncoating of the vector capsid in close vicinity to the host nuclear membrane and promotes vector DNA translocation through nuclear pores, thus strongly stimulating gene transfer in both dividing and non-dividing cells [11]. In its 3′ untranslated region, the transfer plasmid harbors the woodchuck hepatitis virus posttranscriptional regulatory element (WPRE), a *cis*-acting regulatory module that enhances transgene expression by increasing mRNA export and stability [12]. The genes included in the transfer plasmid are flanked by 5′ and 3′ long-terminal repeats [13]. The sequences of the 5′ and 3′ LTRs are strictly identical but have two distinct roles during a viral cycle. The 5′ LTR is crucial for synthesis of the vector RNA through its promoter region named U3. During the reverse transcription step, the template to regenerate the U3 region of the two LTRs is derived solely from the 3′ LTR region. Thus, after LV reverse transcription, the U3 deletion is duplicated from one end to the other of the vector genome, leading to a vector genome completely devoid of HIV-1-based promoter and transcriptional regulatory sequences. This safety lock is the basis of the “self-inactivating” property of LVs [14] (Figure 1).

Lentiviruses and LVs gain access to the host cell either through direct membrane fusion or receptor-mediated endocytosis via the binding of envelope glycoproteins to their receptors. After initial fusion with the host cells, the viral core is transported to the nucleus, as follows. After docking to the nuclear pore, the reverse transcription of ssRNA to dsDNA occurs and is accompanied by the formation of the DNA flap. At this step the uncoating and release of the viral genome from the capsid allows the pre-integration complex to cross the nuclear pore. In the case of lentiviruses or integrative LVs, following access to the host nucleus, the pre-integration complex integrates into the chromosome through the action of the integrase. In non-integrative LVs, in which the catalytic site of the integrase is mutated, the viral DNA cannot integrate into the host chromosome and is maintained in the nucleus as an episomal non-integrated form [11,15]. The lack of integration does not reduce the ability to induce an effective immune response and only requires adjustment of the vector dose [4,16]. Practically, the same immunization efficacy can be achieved with 10 times more non-integrative than integrative LVs.

Regarding the size of the insert that can be incorporated into the genome of these vectors, given the size of the HIV genome (9.1 kb) and that of the LV genome (2.6 kb), there is theoretical space for a 6.5 kb long insert in the LV genome before it exceeds the size of the native viral genome and impacts intrinsic packaging capabilities. However, it is accepted that for inserts longer than 3 kb, the LV production titer undergoes a semi-logarithmic reduction of 3 times per kb [17]. Therefore, it is necessary to define the maximum size of a viable insert based on the needs of the vector. For clinical applications which require high production yields, an insert size of 3–4 kb is usually optimal, which allows the inclusion of a genetic message encoding large (poly)antigenic fragments. In comparison, the first, second and third generation of replication-deficient adenoviral vectors can, respe carry inserts of 8, 14 and 37 kb [18], while for adeno-associated vectors (AAV) the insert size is usually limited to 5.2 kb [19]. Modified vaccinia virus Ankara vectors can carry inserts of up to 25 kb [20]. Although LVs are not the viral vectors that can incorporate the longest inserts, they have many other major advantages that are developed along this review.

## 3. Tropism of Lentiviral Vectors for Dendritic Cells

LVs can transduce murine plasmacytoid, myeloid, lymphoid, or bone-marrow-derived DCs and human plasmacytoid, myeloid, or monocyte-derived DCs in vitro [21,22,23]. In humans, the plasmacytoid DC subset is the least and monocyte derived DCs the most susceptible to LV transduction. It is noteworthy that the VSV-G pseudotyping of LVs generates a particular in vivo tropism for DCs, even though transduction of mesenchymal stromal cells, fibroblasts, and myoblasts cannot be excluded. Cell surface heparan sulfate has been proposed as a receptor for the attachment of VSV-G-pseudotyped LVs to the membrane of target cells [24]. However, the low density lipoprotein receptor (LDL-R), expressed at the surface of many cell types, has now been demonstrated to be the main receptor for VSV-G and the major entry port of VSV-G-pseudotyped LV into both human and murine cells [25,26]. Crystal structures of VSV-G interacting with distinct cysteine-rich domains of the LDL-R, established the binding sites of VSV-G and their deletion abolished VSV infectivity, indicating that all VSV-G receptors are members of the LDL-R family [27]. In contrast to the high potential of DCs and myeloid cells to be transduced by VSV-G-pseudotyped LVs, resting T and B lymphocytes and hematopoietic stem cells cannot be transduced by these vectors because they lack LDL-R expression [28].

LVs are the only members of the retroviridae family able to efficiently transduce cells regardless of their dividing or non-dividing status [23,29,30]. Therefore, LVs are suitable for the transduction of all types of antigen-presenting cells, including those that are not in an active proliferation phase, such as monocyte-derived DCs [31]. This property is crucial for vaccine development and results from the fact that the genome of lentiviruses and LVs can be imported into the nucleus, independently of the mitotic status of the host cells. In net contrast, other retroviral vectors transduce only dividing cells because their genetic material is only able to cross the nuclear membrane of mitotic cells [32].

Using an LV harboring the gene encoding for green fluorescent protein (GFP), under the transcriptional regulation of the human β2-microglobulin promoter, we recently demonstrated that LV efficiently transduces all main DC subsets in vivo [7]. Indeed, following intramuscular immunization of mice with this vector, readily quantifiable proportions of CD11c^+^ CD11b^+^ CD8^−^ myeloid, CD11c^+^ CD11b^−^ CD8^+^ lymphoid, and CD11c^int^ B220^+^ plasmacytoid DC subtypes in the local lymph nodes, expressed significant levels of GFP, determined cytometrically at day 5 post immunization. Much smaller proportions of GFP^+^ cells were detected in the other immune cell populations, i.e., macrophages, lymphocytes and neutrophils, harvested from the same anatomical locations. In addition, following immunization of mice with an LV encoding a model antigen under the human β2-microglobulin promoter, among all the above-mentioned cell types, only DCs were able to activate the expansion of naïve antigen-specific T cells [7].

Based on the non-cytopathic and very weak inflammatory properties of LVs, we recently proposed their use in mucosal vaccination, via the nasal route, against the pulmonary pathogens SARS-CoV-2 [4,33,34,35] and *Mycobacterium tuberculosis* [8,9]. Intranasal administration of an LV encoding GFP, under the control of the P_CMV_ or human β2-microglobulin promoter, resulted in GFP expression in only mucosal lung DCs [9,35]. Importantly, when administered intranasally, even at the high doses of 5 × 10^8^ or 1 × 10^9^ TU/mouse, LVs did not modify the composition or profile of the lung innate immune cells, i.e., CD103^+^ DCs, or alveolar or monocyte-derived DCs, alveolar or interstitial macrophages [9], underscoring the weak potential of LVs to induce mucosal inflammation and local immune cell infiltration, which is a remarkable safety property for a vaccine vector.

The potential of LV to target DCs in vivo, which possess the unique property of activating naïve T cells, can explain the robust immunogenicity of LVs. The transgenic antigen produced endogenously by LV-transduced DCs readily enters the antigen-presentation pathway of the MHC-I machinery. In comparison, adenoviral vaccinal vectors target epithelial cells, but not DCs. The transduced epithelial cells produce the transgenic antigen which is released and must be taken up by by-stander DCs to present the antigen to specific T cells. This indirect presentation causes a sizable decrease in the efficacy of antigen presentation and T-cell activation. It is noteworthy that to achieve the same degree of T-cell immune responses, 10^4^ to 10^5^ times less active LV viral particles are required than adenoviral active viral particles [4]. LVs generate longer-lasting and higher-quality cellular immunity and more central memory T cells than an adenoviral (Ad5)-based vector [7].

In addition to the natural tropism of VSV-G-pseudotyped LVs for DCs, the efficacy of LV-based vaccines could be enhanced by their transductional or transcriptional targeting to antigen-presenting cells, which could also minimize off-target effects. We recently reviewed these approaches in detail [4]. Briefly, transductional approaches are based on LV pseudotyping with glycoproteins other than VSV-G. For example, pseudotyping of LVs with measles virus glycoprotein (MVG) confers tropism towards CD46^+^ cells and otherwise promotes fusion of the virus and host cell membranes [36,37]. However, MVG pseudotyping makes LVs targets of pre-existing immunity to measles virus that is strong and very common in human populations [38]. LV pseudotyping with a mutated sindbis virus envelope glycoprotein (SVG) [39,40] increases tropism for cells expressing heparan sulfate proteoglycan or DC-specific ICAM-3-grabbing nonintegrin (DC-SIGN). Immunization with such LVs increases DC transduction and improves immunogenicity [39,40]. Another approach is the incorporation, into the LV envelope, of nanobodies specific for the surface DC receptors DC2.1 and DC1.8. This strategy has established the feasibility of selective DC transduction. However, the immunogenicity of these LVs did not reach that of conventional VSV-G pseudotyped LVs [36,41].

Transcriptional targeting of LVs relies on the use of specific promoters to favor expression of transgenic antigen by DCs. The pCMV, spleen focus-forming virus (SFFV) and human phospho-glycerate kinase (PGK) promoters are the most commonly used in LVs. However, these promoters are strong, nonselective and constitutive [42]. Among promoters more specifically active in antigen presenting cells, the DC-specific dectin-2 promoter, has been used to induce robust T-cell responses against the melanoma antigen New York esophageal squamous cell carcinoma 1 (NY-ESO-1) [43]. We recently described an LV harboring the human β2-microglobulin promoter, which allows predominant transgene expression in immune cells, notably DCs [7]. Furthermore, the human β2-microglobulin promoter possesses minimal proximal enhancers, which improves the biosafety of the vector. The human β2-microglobulin promoter contains the following highly conserved cis-regulatory elements: (i) interferon (IFN)-stimulated response elements (ISREs), which are the binding sites for the regulatory factors of the IFN family, and (ii) SXY modules, which interact with a multiprotein complex. Both of these elements are regulated by pro-inflammatory immune mediators, which are highly upregulated during the DC-activation process [44,45].

In addition to the tropism of LVs for DCs, the transductional and transcriptional DC targeting approaches, the product encoded by these vectors can be designed to be secreted and specifically addressed to by-stander DCs. We recently developed such LVs which are detailed in Section 6.4 of this review.

## 4. Weak Inflammatory Properties of Lentiviral Vectors

The combined capacity of LV to induce strong and long-lasting T-cell immunity, without being inflammatory is a unique asset for their future use in vaccination and immuno-therapy [16,46,47,48,49]. Indeed, although LVs target DCs in vivo, these vectors generally barely activate DCs [9,50], which largely contributes to their biosafety. Nevertheless, in mice, myeloid DCs produce IFN-I and TNF following their interaction with LVs and through Toll-like receptor (TLR) 3 and TLR7 signaling. Thus, mice genetically deficient for TLR3 and TLR7 signaling were shown to mount weaker T-cell responses to LV immunization than their wildtype counterparts [21]. In human plasmacytoid DCs, the ssRNA genome of LVs can stimulate innate intracellular pathways, resulting in the production of IFN-I and TNF, which activate bystander myeloid DCs [51,52]. Even when injected intravenously, LVs induce only moderate phenotypic and functional DC maturation and inflammatory responses [50]. An in vitro study reported a certain degree of LV-induced DC maturation attributed to VSV-G [53]. In our hands, incubation of murine bone-marrow derived DCs with a pre-GMP quality LV displayed only minor CD86 upregulation and moderate increases in the proportion of MHC-I^hi^ or -II^hi^ cells, even at a high multiplicity of infection. These DCs secreted minute levels of IFN-α, IFN-β, CCL5, and IL-10 but did not secrete IL-1α, IL-1β, IL-6, or TNF, which are the main inflammatory mediators secreted by maturing DCs [9].

In addition to their intrinsically weak inflammatory characteristics, another attractive property of LVs is the absence of pre-existing immunity in human populations against these vectors. The presence of pre-existing humoral or T-cell vector immunity limits the efficiency of antigen-presenting cell transduction by their rapid elimination. This is a particular limitation in the use of adenoviral and herpes simplex virus type 1 vectors, which show high seroprevalence in the human population [54]. Pre-existing immunity against the heterologous LV envelope VSV-G is rarely found in human populations because of rare pre-exposition to VSV [26].

It is noteworthy that even though the LV particles are not toxic and scarcely inflammatory, in LV batches, residual host cell proteins, residual nucleases used in the purification steps, residual bovine serum albumin, residual nucleic acids and endotoxins could be source of undesirable toxicity or inflammation. We never detected in vitro or in vivo toxic potential or undesirable inflammatory properties in many LV preparations from our laboratory or in good manufacturing practice (GMP) batches that we have been able to test so far. The concentration of such contaminants not to be exceeded in viral vector GMP batches is not predefined in the regulatory FDA recommendations [55]. It depends on each vector preparation and detection method, and it is especially necessary that the safety of viral vector GMP batches be demonstrated indetailed and meticulous pre-clinical toxicology assays, according to the FDA recommendations. It is nowadays possible for industrial partners to produce LV GMP batches with very low levels of contaminants and which have obtained the regulatory authorizations to be administered in humans [56].

## 5. Biosafety of Lentiviral Vectors

One main characteristic of LVs used in vaccination and immuno-oncotherapy is their non-integrative property. Unlike murine leukemia virus (MLV)-based γ-retroviral vectors, no serious genotoxicity has thus far been observed with integrative LV-based gene therapy clinical studies performed to correct primary immunological genetic deficiencies [57,58,59]. It is well established that the integration sites of integrative LVs do not favor the activation of proto-oncogenes and that they are a safer choice for clinical use. Nevertheless, to avoid any risk of insertional mutagenesis, nonintegrative LVs have been developed. These LVs harbor a missense D64V mutation in the catalytic triad of their integrase, which prevents integration of the vector DNA into the host chromosome [60]. In the absence of integration, the viral DNA remains in a circular episomal form, mostly in the form of 1LTR circles, which are effective for gene expression [61]. As mentioned above, nonintegrative LVs remain largely immunogenic and only require adjustment of the vector dose [4,16].

Even though the LVs to be used for vaccination or immuno-oncotherapy are nonintegrative, the demonstrated biosafety of the integrative LVs used in gene therapy only reinforces the non-genotoxicity of these vectors and *a fortiori* that of nonintegrative vectors. Gene therapy via viral vectors is based on (i) γ-retroviruses, (ii) lentiviruses, (iii) adenoviruses, and (iv) adeno-associated viruses. Among them, γ-retroviruses and lentiviruses are retroviruses, meaning that their RNA genome is converted into double stranded DNA in the transduced cell by reverse transcriptase. It is noteworthy that the cases of activation of proto-oncogenes and associated T-cell acute lymphoblastic leukemia (T-ALL) and acute myeloid leukemia (AML) recorded following gene therapy of X-linked severe combined immunodeficiency disease (SCID) resulted from the use of a murine leukemia virus (MLV)-based γ-retroviral vector [54,62] but not LVs. In stark contrast to MLV, no serious adverse effects have been observed with LV-mediated clinical studies [59]. Comparative analysis of the integration sites of the genetic material of γ-retroviruses and lentiviruses showed that integrative LVs preferentially integrate near transcriptional units and do not favor the activation of proto-oncogenes, despite billions of integration sites. On the contrary, γ-retroviral vectors preferably integrate near transcriptional start sites [59] and the enhancers of proto-oncogenes [63]. Importantly, very recent reports on the long-term (4 to 9 years) biosafety and efficacy of LV-based hematopoietic stem/progenitor cell gene therapy for Wiskott-Aldrich syndrome phase I/II [57] and long-term outcomes of LV-based gene therapy for β-hemoglobinopathies [58] show an excellent record of biosafety, with no proto-oncogene activation or clonal cell expansion. Even the use of highly optimized integrative LVs for gene therapy, with new enhancers that greatly increase the number of inserted gene copies, has shown no increase in integrative mutagenesis in hematopoietic stem cell progenitors [64,65]. Finally, in more than 35 years of research and follow-up of cohorts of AIDS patients, HIV has never been reported to be associated with insertional activation of proto-oncogenes. In parallel, integrative LVs are also FDA approved and have been successfully used to deliver genes encoding chimeric antigen receptors (CARs) into mature T cells for cancer immunotherapy.

A phase I/IIa clinical study based on therapeutic HIV treatment with a non-replicative, self-inactivating, integrative LV has so far established the immunogenicity and short- and long-term (five years) biosafety and tolerability of this vector in a total of 38 patients [56]. A nonreplicative, self-inactivating, nonintegrative LV has also been evaluated in a phase I/IIa clinical trial as a therapeutic vaccine candidate for patients with sarcoma and other solid tumors expressing New York esophageal squamous cell carcinoma-1 (NY-ESO-1). Only grade 1 or 2 treatment-related adverse events were reported and no ≥3 treatment-emergent adverse events. No genotoxicity has been observed and good quality T-cell responses have been successfully detected and shown to be associated with a potential clinical benefit [66,67]. Overall, the cumulative results from numerous clinical observations in gene therapy and immunotherapy in humans are highly reinsuring in terms of the biosafety of LVs.

## 6. Lentiviral Vector-Based Vaccination in Preclinical Models

LV-based vaccination against infectious diseases and malignancies have consistently demonstrated the induction of strong humoral and cellular immune responses, accompanied by highly significant protection in preclinical animal models. In a recent review, we have exhaustively described the results of immunogenicity/protection in pre-clinical and clinical studies already obtained with the use of LVs in the vaccination against infectious diseases and immuno-oncotherapy [4]. In particular, we can mention the use of LVs by our group and others in: (i) treatment of melanoma by targeting Melan-A, NY-ESO-1, Hsp70, tyrosine related protein 2 (TRP2), tyrosine related protein 1 (TRP1), human telomerase reverse transcriptase (hTERT), melanoma antigen gp100 [39,43,67,68,69,70,71,72,73,74,75,76,77,78,79,80], (ii) treatment of prostate cancer by targeting prostate stem cell antigen (PSCA) [81], and (iii) vaccination against infectious agents such as hepatitis B virus (HBV) [82], hepatitis C virus (HCV) [83], human papilloma virus [84], influenza virus [48] and HIV or SIV [46,47,49,85,86,87,88,89]. Information established by our team in preclinical proof of concept studies on the efficacy of LV-based vaccination/immuno-oncotherapy is summarized in Table 1. Below, we discuss, in a non-exhaustive manner, our most recent results generated in preclinical animal infectious models.

### 6.1. LV-Based Vaccine Candidates against Flaviviruses

Viruses that belong to the family Flaviviridae are most commonly transmitted to humans through the bite of an infected mosquito or tick and are responsible for over 400 million cases of human disease annually, mainly in tropical and sub-tropical areas [96]. The Flaviviridae family includes a range of important human pathogens, such as dengue (DENV), West Nile (WNV), Japanese encephalitis (JEV), Zika (ZIKV) and yellow fever (YFV) viruses. Viral particles are surrounded by a lipid envelope that contains a single-stranded positive-sense RNA genome encoding three structural proteins: capsid (C), pre-membrane/membrane (prM/M), and envelope (E), and seven nonstructural proteins, NS1, NS2A, NS2B, NS3, NS4A, NS4B, and NS5. Several preclinical studies have demonstrated that LVs could be an efficient platform for the development of vaccines against different flaviviruses. A single administration of a nonintegrative LV encoding a secreted form of the E protein of WNV induced a robust and long-lasting B-cell response just a week post-immunization and protected mice against lethal WNV challenge [29,49]. Co-expression of the surface proteins prM and E of JEV by an LV elicited high titers of anti-JEV antibodies that demonstrated high neutralization activity against the G1, G3, and G5 genotypes of JEV in mice and pigs [91,92].

In 2016, an epidemic caused by a new variant of the Zika virus (ZIKV) posed a threat to public health around the world. For the first time, infection with the virus was linked to the development of neurological diseases in adults and congenital syndromes in the fetuses of pregnant women [97]. As no effective drug to treat the disease was available, the development of a vaccine capable of thwarting the spread of the disease and preventing the risk of re-emergence was desirable. The most common mechanism of protection against viral infection is blocking the transmission of infectious viral particles by neutralizing antibodies. Given our previous results obtained with WNV and JEV, we sought to induce such protection by designing LVs that express the structural proteins of ZIKV. The antigens were based on a consensus sequence of circulating ZIKV strains to provide the largest possible coverage for such a LV vaccine [93]. Two candidate LV vaccines were tested, the first encoding prM followed by the full-length envelope (prM-E) and another that encoded the soluble E protein. Both LV constructs induced robust humoral responses against E protein in mice (Figure 2A). However, only the LV prM-E construct was able to elicit antibodies that efficiently cross-neutralized two ZIKV strains (Figure 2B). Vaccination of IFNα/βR°/° A129 mice, which are highly susceptible to ZIKV, fully protected them from morbidity and mortality after a lethal challenge. Indeed, neither viremia nor detectable infectious viral particles were detectable in the brain or testis, which suggests that a LV-based vaccine could also prevent sexual transmission (Figure 2C). After a single LV injection, protection was readily effective as early as one week and lasted up to six months. Although the genetic information transferred by LVs does not integrate into the host genome, the persistence of antigen production and/or release of prM-E virus-like particles could potentially enhance the affinity, avidity, and anti-pathogen activity of anti-prM-E antibodies, leading to a strong and increasing neutralizing antibody response over time. Generating an effective and rapid protective response through a single vaccination is clearly an asset for an emergency Zika vaccine. The LV-based vaccine with its single-dose format and no additional adjuvant formulation is of particular interest. Indeed, many regions affected by ZIKV present significant healthcare barriers. Booster vaccination of these hard-to-reach and difficult-to-follow susceptible populations remains highly uncertain. With its single-dose format, the LV-based vaccine would allow more people to be vaccinated while providing them with full protection against Zika disease. Several Zika vaccine candidates have shown promising results in controlling ZIKV infection but they required adjuvant and multiple-dose administration, therefore, falling short in addressing the aforementioned constraints [98].

Of note, the risk of antibody-dependent enhancement (ADE), which would have devastating effects upon a subsequent encounter with viruses closely related to ZIKV, is of concern. Given such a potential risk, our team is working on a T cell-based vaccine to target and clear ZIKV via a T cell-mediated route, independent of the antibody neutralizing paradigm.

Dengue virus (DENV) is responsible for an estimated 390 million human infections and up to 100 million cases of dengue fever per year, making this virus a major threat to public health worldwide [99]. A particular difficulty related to the development of efficient vaccines against DENV is that the human disease is caused by four distinct serotypes of DENV that co-circulate in tropical and sub-tropical regions. Initial infection with one DENV serotype provides long-term protection against that serotype but only short-term cross-protection against the others [100]. Re-infection with a heterologous serotype increases the risk of more severe forms of the disease, such as dengue hemorrhagic fever (DHF) and dengue shock syndrome (DS). The most prevalent theory to explain such disease enhancement, known as ADE, is that antibodies produced in response to the first infection are not able to efficiently neutralize other DENV serotypes, but instead bind to viral particles and facilitate their uptake by antigen-presenting cells, which are the natural targets of DENV. This scenario would lead to increased viral replication in the host and, ultimately, to more severe disease [101]. Knowledge of ADE led to the understanding that an efficient vaccine against DENV should simultaneously protect against all four serotypes. Another hypothesis that was put forward to explain the increased severity of heterologous DENV infections is called “original antigenic sin” and postulates that T-cell responses formed after infection with the first DENV serotype may not efficiently kill the host cells infected with another serotype [102]. Nevertheless, such T cells could produce high amounts of pro-inflammatory cytokines, leading to a dysregulated immune response and more severe disease. In addition, the initial expansion of such poorly specific T cells could delay the expansion of more specific T-cell clones that could efficiently eliminate virus-infected cells [102]. Although both theories of DENV pathogenesis were initially considered, a number of recent studies have suggested that T-cell responses play a protective role against DENV infections rather than contributing to its pathogenesis [101]. Humans with a history of asymptomatic DENV infections have strong T-cell responses, mainly directed against conserved nonstructural proteins of DENV (NS3, NS5, and NS4B) [103]. In addition, studies in preclinical models demonstrated that T-cell responses are essential for protection against heterologous DENV infections and can prevent DENV-induced ADE [104,105]. Heterologous DENV infections also redirected T-cell responses to the antigens encoded by conserved regions of the genomes of different DENV serotypes. As people who have been repeatedly infected by various DENV serotypes often ultimately develop long-lasting immunity to DENV, targeting of the conserved epitopes located in the NS proteins by T-cell responses could be considered as a correlate of long-lasting protection against DENV disease. It is also worth mentioning that the neutralizing antibody response against DENV does not appear to be a reliable correlate of protection [106], again suggesting an important role of the T-cell response. The above-mentioned studies led to the conclusion that optimal induction of the cross-reactive T-cell response could serve as a principle for creating a “T-cell-based” vaccine that could avoid potential problems associated with the antibody response and simultaneously protect against all serotypes of DENV and potentially other flaviviruses of public interest, such as ZIKV and yellow fever virus. The advantages of a LV-based vaccine platform are expected to maximize the potential of such an anti-DENV T-cell vaccine directed against several flaviviruses. Such vaccine candidates are currently undergoing pre-clinical trials in our laboratory.

### 6.2. LV-Based COVID-19 Vaccine Candidate

Very early in the coronavirus disease 2019 (COVID-19) pandemic, we generated an LV-based vaccine candidate that encodes the full-length sequence of the SARS-CoV-2 spike antigen (LV::S) [34]. This vector elicited high titers of neutralizing antibodies, comparable to those detected in convalescent humans. Importantly, in parallel, LV::S induced a strong and poly-specific CD8^+^ T-cell response against numerous T-cell epitopes of the spike protein. At the early stage of the COVID-19 pandemic, when mice transgenic for the SARS-CoV-2 receptor, human angiotensin converting enzyme (hACE)2, were not yet available, we developed a murine model in which the expression of hACE2 was induced by the transduction of respiratory tract cells through i.n. instillation of an adenoviral (Ad5) vector encoding hACE2 to evaluate the protective potential of LV::S. Such pre-treatment induced the transitory yet strong expression of hACE2 in the lungs and allowed substantial SARS-CoV-2 replication. In this murine model, an i.m. prime followed by an i.n. boost with LV::S triggered a mucosal immune response in the respiratory tract that resulted in full SARS-CoV-2 clearance, prevented undesirable local immune infiltration and inflammation, and protected against injury of the lung parenchyma (Figure 3). In addition, in the highly susceptible Golden hamster, naturally permissive to SARS-CoV-2 replication, a LV::S i.m. prime followed by an i.n. boost also achieved full SARS-CoV-2 clearance, largely reduced inflammation, and prevented deleterious lung infiltration, severe alteration of the bronchiolar epithelium, and interstitial syndromes [34,95]. Since then, numerous expert teams have demonstrated that COVID-19 vaccination by the nasal route induces mucosal, humoral, and cellular immunity at the entry point of SARS-CoV-2 into the host organism. Indeed, it is now accepted that the most effective immunization approach for reducing SARS-CoV-2 transmission is i.n. vaccination [107,108,109,110]. Because LVs are noncytopathic, nonreplicative, and scarcely inflammatory, they are particularly favorable for use in mucosal i.n. vaccination. In stark contrast to LVs, mucosal administration of adenoviral vectors, which are pro-inflammatory and target pre-existing immunity [4], is risky and mRNA vaccines, which are packaged in poly-ethylene-glycol-containing nanoparticles, cannot be used for mucosal administration in their current formulation [111].

Although the neurotropism of SARS-CoV-2 is well established [112,113,114], the COVID-19 vaccine strategies developed thus far have not taken into account protection of the brain or central nervous system. We generated a transgenic mouse strain (B6.K18-hACE2^IP-THV^) expressing hACE2 under the control of the keratin 18 promoter, which showed strong *hACE2* mRNA expression in the lungs, as well as in the brain, together with strong permissiveness to SARS-CoV-2 replication in both pulmonary and cerebral anatomical sites [33]. This preclinical model is highly susceptible to SARS-CoV-2, as inoculated mice die in 4 to 5 days in the absence of treatment/vaccination. In this very stringent model, we demonstrated that an i.m. prime followed by an i.n. boost with LV::S led to the full clearance of SARS-CoV-2 at both anatomical sites (Figure 4). Importantly, protection of the central nervous system required an i.n. LV::S boost, which correlated with the presence of CD8^+^ T cells in the olfactive bulbs, quantifiable by both cytometry and immunohistochemistry [33].

These observations also established the strong cross-protective potential of LV::S, despite the significant reduction in cross-sero-neutralization capacity of the induced antibodies against mutated spike. Indeed, the immune mechanism involved in the broad spectrum protection of LV::S is based on robust CD8^+^ T-cell immunity. Importantly, µMT KO mice, which are devoid of the transmembrane portion of the heavy Ig µ chain and, thus, deficient in mature B-cell subsets and antibodies, were still highly significantly protected by LV::S. The T-cell immune response was still highly effective against emerging SARS-CoV-2 variants of concern, which accumulate amino acid substitutions in spike, notably in the receptor binding domain (RBD), to escape the neutralizing antibodies generated by previous vaccination or infections. In stark contrast to the viral escape from antibodies, only a very limited number of spike T-cell epitopes have been reported to be altered or lost because of spike mutations. For an infected cell to efficiently escape from T-cell recognition, the anchor residues of a viral T-cell epitope have to be altered/lost, which has been shown to barely occur. Moreover, a non-anchor amino acid substitution inside a T-cell epitope does not necessarily preclude its recognition by TCRs of a polyclonal T-cell response. In addition, spike is a largely multi-T-cell epitopic antigen and even if one of its T-cell epitopes is completely lost, other epitopes remain detectable by the poly-specific TCR repertoire induced by a powerful T-cell vaccine, such as LV::S [33].

It is now widely accepted that the adaptive immunity initially induced by the first-generation COVID-19 vaccines wanes, and needs to be boosted and its specificity broadened. Therefore, at a more advanced stage of the pandemic, when variants of concern were emerging incessantly and human populations began to be massively vaccinated with the first-generation vaccines, we set up an optimized cross-protective i.n. booster study against COVID-19 by generating an LV encoding the full-length sequence of spike from the SARS-CoV-2 Beta variant, stabilized by K^986^P and V^987^P substitutions in the S2 domain (LV::S_Beta-2P_) [35]. Indeed, LV::S_Beta-2P_ generated the best cross-reactive antibody responses against the other variants of concern. Mice primed and boosted i.m. with an mRNA vaccine, with waning primary humoral immunity at four months after vaccination, were boosted i.n. with LV::S_Beta-2P_. A strong boost effect on the waning mRNA-induced immunity was detected through cross-sero-neutralizing activity and systemic T-cell immunity. More importantly, mucosal anti-spike IgA, lung-resident B cells, lung effector memory T cells, and lung resident T cells were efficiently induced only after the i.n. boost with LV::S_Beta-2P,_ but not following a third dose of the mRNA vaccine. In addition these immune responses correlated with complete pulmonary protection against the SARS-CoV-2 Delta variant, demonstrating the suitability of the LV::S_Beta-2P_ vaccine candidate as an i.n. booster against COVID-19. LV::S_Beta-2P_ vaccination was also fully protective against Omicron infection of the lung and central nervous system in highly susceptible B6.K18-hACE2^IP-THV^ transgenic mice [35].

### 6.3. LV-Based Tuberculosis Vaccine Candidate

According to the World Health Organization, a quarter of the world’s population is infected with *Mycobacterium tuberculosis* and 5 to 10% of infected people will develop active tuberculosis [115]. In 2021, 10.6 million people developed active tuberculosis, which represents an increase of 4.5% from 2020, and 1.6 million people died from the disease. The burden of drug-resistant tuberculosis also increased by 3% between 2020 and 2021. As a particularly severe impact of the COVID-19 pandemic, an increase in the incidence of active and drug-resistant tuberculosis was observed for the first time since 2005. The only vaccine currently on the market is *Mycobacterium bovis* Bacillus Calmette-Guérin (BCG), which is not sufficiently effective to have an epidemiological impact. Even though LVs are highly efficient at inducing CD8^+^ T-cell responses, LV-encoded antigens do not efficiently enter the endosomal MHC-II presentation pathway. As CD4^+^ T cells present the best correlate of protection against intracellular bacteria such as *M. tuberculosis* [116,117], inefficient induction of CD4^+^ T-cell response by viral vectors significantly limits effective protection against such pathogens.

To optimize LVs for CD4^+^ T-cell induction, we extended antigens at their N-terminal end with the MHC-II-associated light invariant chain (li) [118], which contains an endosome-routing signal sequence [9]. We first established that this modification oriented the (poly)antigens into the MHC-II pathway of LV-transduced DCs in vitro, without reducing their routing to the MHC-I presentation pathway. In addition, mice immunized i.m. or i.n. with such an optimized LV-encoded mycobacterial poly-antigen mounted systemic and mucosal polyfunctional CD4^+^ and CD8^+^ T-cell responses. Prophylactic vaccination by a systemic prime followed by an i.n. boost with the optimized poly-antigenic LV largely protected the lungs against a challenge with virulent *M. tuberculosis* and induced a marked reduction in the lung mycobacterial burden in the murine model [9].

### 6.4. LV Encoding Secreted Protein Cargo That Targets Dendritic Cells to Induce Anti-Mycobacterial Immunity

We recently developed a multifunctional LV that, relative to conventional LVs, is optimized to: (i) facilitate poly-antigen delivery, (ii) target secreted antigens to DCs and activate them, (iii) target antigens into the MHC-II machinery, and (iv) in addition to CD8^+^ T cells, also activate polyfunctional CD4^+^ T cells [8]. This approach is based on the generation of LVs encoding engineered C-type lectins, i.e., collectins, as scaffolds bearing multiple antigens, which are able to spontaneously self-assemble and be secreted by the initially LV-transduced DCs. Collectins are collagen-containing multimeric macromolecules that recognize surface lipids or oligosaccharides of microorganisms [119]. Among collectins, mannan-binding lectin (MBL) and *surfactant-associated* protein D (SPD) are composed of four domains: (i) an N-terminal cysteine-rich crosslinking domain, (ii) a collagen-rich domain, (iii) an α-helical neck, and (iv) a carbohydrate-recognition domain. We used MBL and SPD as antigen carriers [120]. The MBL and SPD monomers are able to spontaneously self-assemble into helicoidal trimers (Figure 5A). The trimers then tetra- or hexamerize to form macromolecules. We generated LVs encoding collectin monomers engineered by replacing substantial portions of their collagen-rich domain by *M. tuberculosis* poly-antigens (Figure 5B). Furthermore, to increase their immunogenicity, these macromolecule carriers were also engineered to be directly delivered to DCs subsequent to their release by the initially transduced cells via the co-stimulatory molecule CD40 [121,122,123]. This was accomplished by replacing their C-terminal carbohydrate-recognition domain by the ectodomain of the CD40 ligand (CD40L) [121,122]. The spontaneous trimerization of these scaffolds leads to CD40L homo-trimerization, necessary for productive interactions with CD40. In this approach, the macromolecules are secreted by the initially LV-transduced DCs and taken up by bystander DCs. The secreted macromolecules are able to activate by-stander DCs, which can exhibit slight phenotypic and functional maturation (Figure 5C). The macromolecules taken up by the bystander DCs readily reach the endosomal MHC-II pathway and induce marked MHC-II antigen presentation. When used to vaccinate mice against mycobacteria antigens, such LVs induce both CD8^+^ and CD4^+^ T-cell responses at the systemic and pulmonary levels (Figure 5D). Such vectors also conferred a significant protective booster effect (Figure 5E), consistent with the importance of CD4^+^ T cells for protection against *M. tuberculosis* [116,117]. Given the instrumental contribution of CD4^+^ T cells in orchestrating innate and adaptive immunity, this strategy can have a broad range of applications in the vaccinology field [8].

## 7. Concluding Remarks

Compared to other viral vectors, the LV platform provides a highly effective vaccination vector with several advantages, including, notably, a high potential for transducing DCs in vivo. The resulting endogenous expression of transgenic antigens inside these most potent antigen-presenting cells and direct access of transgenic antigens to the presentation pathways without the need for external antigen capture or cross-presentation are key to the efficiency of LVs in inducing T-cell responses. In this review, we have mainly summarized the immunological aspects of LVs, their recent optimization to induce CD4^+^ T cells, and our recent data on LV-based vaccination in preclinical models, including prophylaxis against flaviviruses, SARS-CoV-2, and *Mycobacterium tuberculosis*. LVs induce robust, strong, and long-lasting humoral and CD8^+^ T-cell immunity and effective protection in many infectious indications. The absence of pre-existing immunity to LVs in human populations, as well as the very weak pro-inflammatory properties of these vectors, also pave the way for their use in mucosal vaccination. The use of LVs in mass vaccination against infectious diseases could potentially be limited because of their relatively costly and challenging large-scale production. However, we did not identify any major technological bottlenecks that would prevent the development of a cost-effective and up-scalable LV production process in the near future.

## Figures and Tables

**Figure 1 pharmaceutics-15-00846-f001:**
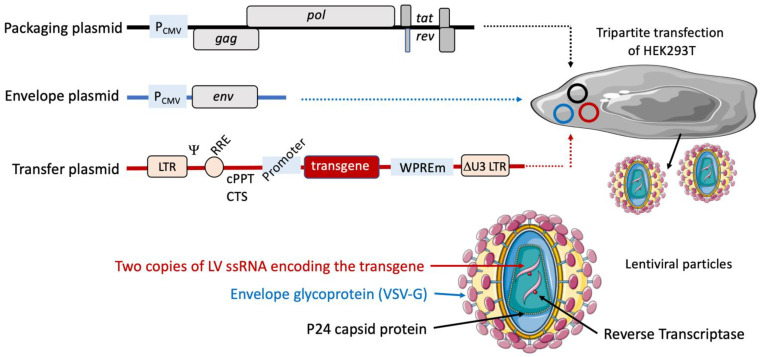
Lentiviral plasmids and the principle of production of lentiviral vectors. The genetic information necessary for the production of LVs is distributed over at least three distinct plasmids. The packaging plasmid contains *gag*, *pol*, *tat*, and *rev* genes. The envelope plasmid encodes VSV-G. These elements are under the transcriptional control of the cytomegalovirus promoter (P_CMV_). The transfer plasmid contains the Ψ packaging signal, RRE, and cPPT/CTS cis-acting elements upstream of the transgene of vaccinal interest under an internal promoter of choice. At its 3′ untranslated region, this plasmid also harbors the mutated WPRE sequence (WPREm), which enhances transgenic antigen expression. In the transfer plasmid, these genes are flanked by two LTRs. LV particles are generated by the tripartite transfection of the HEK293T cell line with the three plasmids. The LV particles are harvested in the culture supernatants at day 3 post-transfection.

**Figure 2 pharmaceutics-15-00846-f002:**
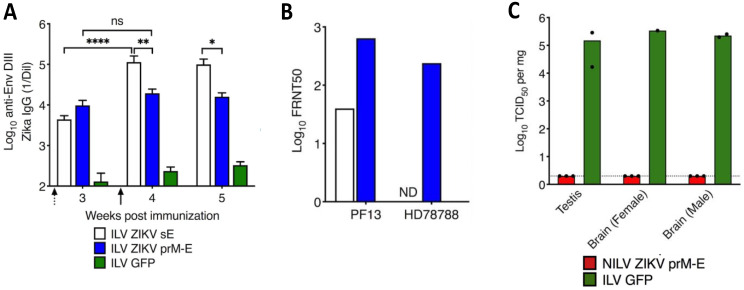
The humoral response induced by LV prM-E vaccine candidate fully protects from lethal ZIKV challenge. C57BL/6 mice were injected i.p. with integrative LV (ILV) ZIKV sE, LV ZIKV prM-E, or LV GFP, as a negative control. (**A**) IgG antibody titers specific to EDIII (a target of protective neutralizing mAbs against ZIKV) were determined by ELISA after priming (dotted arrow) and boosting (black arrow) with the respective LV vaccine candidates(ns = not significant, * *p* > 0.05, ** *p* > 0.01, **** *p* > 0.0001, Mann-Whitney test). (**B**) Serum neutralizing (Focus Reduction Neutralization Test with 50% neutralization cutoff, FRNT50) titers against PF13 and HD78788 ZIKV strains at 4 weeks postimmunization. (**C**) Female or male IFNα/βR°/° A129 mice were immunized with a single dose of non-integrative (NILV) ZIKV prM-E or ILV GFP. Immunized mice were challenged at 4 weeks post-immunization with ZIKV HD78788strain. Viral loads in testis and brain quantified 28 days post-challenge using a TCID50 (Median Tissue Culture Infectious Dose) assay. Each dot represents one mouse. Adapted from [93].

**Figure 3 pharmaceutics-15-00846-f003:**
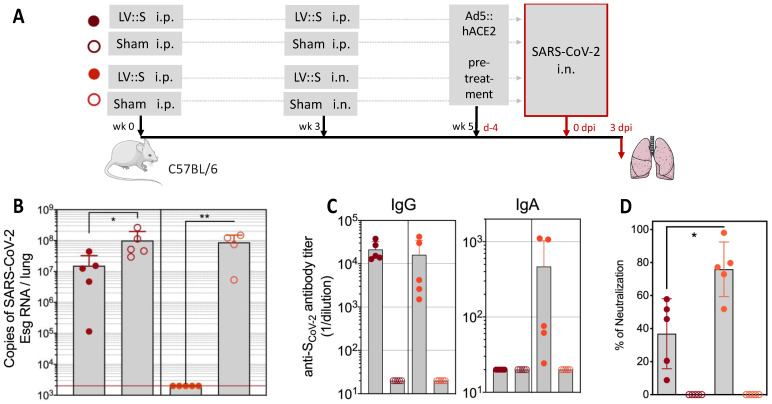
An intranasal boost with LV::S fully protects against SARS-CoV-2 replication. (**A**) Timeline of prime (i.p.) and boost (i.p. or i.n.) with LV::S in C57BL/6 mice, followed by pretreatment with an Ad5::hACE2 vector encoding the human ACE2 receptor four days before an i.n. challenge with SARS-CoV-2. The control LV vector encodes an irrelevant antigen (sham). (**B**) Pulmonary viral content determined at three days post infection (dpi) by qRT-PCR specific for a subgenomic RNA (Esg) expressed only by replicating SARS-CoV-2. The red line indicates the limit of detection. (**C**) Titers of IgG and IgA antibodies specific for SARS-CoV-2 spike antigen, determined by ELISA, in lung homogenates. (**D**) Anti-SARS-CoV-2 antibody neutralizing activity detected in lung homogenates using spike-bearing pseudoviral particles. Statistical significance of the differences was determined by the Mann-Whitney *U* test (* *p* < 0.02, ** *p* < 0.01). Adapted from [34] ).

**Figure 4 pharmaceutics-15-00846-f004:**
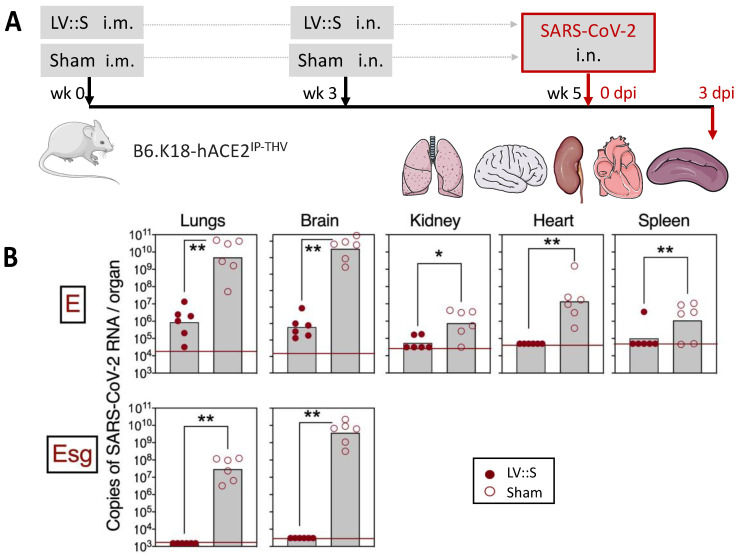
Vaccination with LV::S protects both the lungs and brain against SARS-CoV-2 infection in highly susceptible B6.K18-hACE2^IP-THV^ transgenic mice. (**A**) Timeline of the prime (i.m.)-boost (i.n.) with LV::S and SARS-CoV-2 challenge in B6.K18-hACE2^IP-THV^ transgenic mice. (**B**) Viral RNA content, determined in various organs at 3 dpi by a conventional E-specific or a sub-genomic Esg-specific qRT-PCR. Red lines indicate the detection limits (* *p* > 0.05, ** *p* > 0.01, Mann-Whitney test). Adapted from [33].

**Figure 5 pharmaceutics-15-00846-f005:**
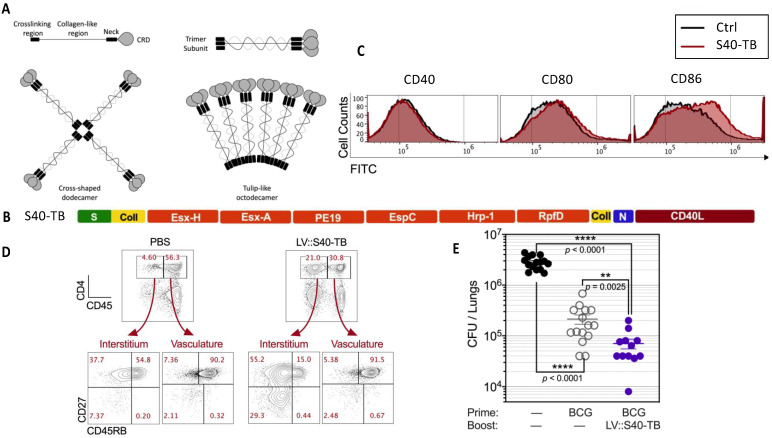
Collectin-based antigen carriers as encoded by LVs and their immunogenicity and protective potential against *M. tuberculosis*.(**A**) Structure of collectin monomers and polymers. CRD = carbohydrate-recognition domain. Self-assembled triple helixes can multimerize to form cross-shaped dodecamers or “tulip bouquet-like” octodecamers. Adapted from [119]. (**B**) Representation of the primary structure of the engineered surfactant D, harboring mycobacterial antigens and substituted within its CRD domain by the ectodomain of CD40L. S = crosslinking region, Coll = collagen-like region, N = neck region. (**C**) Phenotypic maturation of murine bone-marrow-derived DCs, untreated or incubated overnight with supernatants from HEK-293T cells transduced with a control LV or LV::S40-TB. Expression of co-stimulatory molecules on the surface of DCs, as assessed by cytometry. (**D**) Mucosal CD4^+^ T cells induced by i.n. immunization of mice with LV::S40-TB at 14 days post-immunization. Lung CD4^+^ T cells were distinguished by their location in the interstitium (CD45_i.v_^−^) or vasculature (CD45_i.v_^+^) subsequent to i.v. injection of PE-anti-CD45 mAb, 3 min before sacrifice, which allowed determination of the CD27 vs. CD45RB profile of CD4^+^ T cells in the lung parenchyma and vasculature. (**E**) Evaluation of the booster effect of LV::S40-TB in the protection against *M. tuberculosis*. C57BL/6 mice were unvaccinated, vaccinated with BCG alone (s.c., week 0) or BCG-primed (s.c., week 0) and LV::S40-TB-boosted (s.c., week 5 and i.n., week 10) and challenged with the virulent *M. tuberculosis* H37Rv strain (i.n., week 12). Lung mycobacterial loads were determined in week 17. The significance of the differences was determined using the Mann-Whitney *U* test. Ns = not significant. Adapted from [8].

**Table 1 pharmaceutics-15-00846-t001:** Recapitulative of our pre-clinical proofs of concept of the efficacy of LV-based vaccination/immuno-oncotherapy.

Pathology/Virus	Main Results	Reference
West Nile Virus	The single administration of a nonintegrative LV encoding a secreted form of the envelope protein of a virulent strain of West Nile virus induces a robust antibody response, as well as full and long-lasting sterilizing protection from a challenge with a lethal dose of West Nile virus in a murine model.	[29,49]
AIDS	Induction of broad CD8^+^ cytotoxic T-cell responses with a LV encoding HIV-1 polyepitopes in humanized mice. LV-based prime-boost vaccination demonstrating protective immunity against simian immunodeficiency virus SIVmac251 challenge in macaques.	[46,90]
Japanese Encephalitis Virus	An LV encoding the pre-membrane and envelope protein (prME) of Japanese encephalitis elicits a broad neutralizing antibody response and protection in pigs.	[91,92]
Malaria	Long-term sterile protection against malaria in mice immunized with a nonintegrative LV encoding *Plasmodium yoelii* circumsporozoite protein (CSP) and challenged with sporozoites.	[16]
Zika	A single dose of nonintegrative LV-based vaccine candidate provides rapid and durable protection against Zika virus.	[93]
COVID-19	A nonintegrative LV encoding the Spike envelope protein of SARS-CoV-2, used in an i.m. prime dose, followed by an i.n. boost, induces sterilizing protection of the respiratory system and the central nervous system in hACE2 humanized transgenic mice and golden hamsters. The efficacy of this LV-based COVID-19 vaccine candidate, used as a late i.n. booster in initially mRNA-vaccinated individuals, is superior to that of a third mRNA dose, which correlates with its ability to induce mucosal IgA and resident memory B and T cells in the respiratory airways.	[33,34,35,94,95]
Tuberculosis	Development of a new-generation LV-based tuberculosis vaccine/booster candidate. Two new generations of LVs have been generated. Their strong immunogenicity for CD4^+^ T cells correlates with a high protective potential when administered either alone or as a booster in BCG-primed mice.	[8,9]

## Data Availability

Not applicable.

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
