# Peer review of "Lentiviral Vectors as a Vaccine Platform against Infectious Diseases"

_pharmaceutics, 2023, doi:10.3390/pharmaceutics15030846_

Round 1

Reviewer 1 Report

Nemirov et al. review lentiviral vectors (LVs) as a vaccine platform against various infectious diseases with a focus on their potential for transducing dendritic cells in vivo. Additionally, the review discusses several advantages of LVs over adenoviral vectors.

This is an interesting review highlighting the potential benefit of developing LVs to target infectious diseases as they can induce strong, robust, and long-lasting humoral, CD8+ T-cell immunity and effective protection, and have even been optimized to induce CD4+ T cells. Additionally, the review also discusses several important aspects, such as the absence of pre-existing immunity to LVs in humans as well as their weak pro-inflammatory properties, making them good candidates for the development of effective vaccines. The authors have previously carried out several pre-clinical studies to investigate lentiviral vector-based vaccination, including prophylaxis against flaviviruses, SARS-CoV-2, and Mycobacterium tuberculosis, the details of which have also been discussed at length in this review. This is a comprehensive and informative review of various aspects of LVs, but the following minor points need to be addressed to improve the quality of the paper.

A sentence briefly describing the different strategies that are used to transfer genes into the host cell should be added at the onset of the paper (in the introduction section).

Page 1 Lines 9-10: The importance/advantages of LVs can be discussed and emphasized here by comparing the package size of LVs, and other retroviral gene therapies, including adenovirus vectors and adeno-associated virus.

Page 1 Line 11: Before mentioning about integrase, please add a sentence that mentions the limitation of LVs in that they carried a risk of insertional mutagenesis and off target effects and that this limitation was overcome by the use of integrase. Although there is a brief mention of LVs carrying a risk of insertional mutagenesis in the subsequent section, highlighting this aspect in the introduction section would be better to understand the characteristics of LVs.

Page 1 Line 25: Please add 2-3 sentences towards the end of the introduction section that summarize the key points discussed in the review to better understand the rationale of the paper.

Page 1 Line 28: Please rewrite this sentence and provide more details here.

Page 6 Line 31: Abbreviations and their full forms should be written at the first appearance in the text.

Reviewer 2 Report

Manuscript ID: pharmaceutics-2239068

Type of manuscript: Review

Title: Lentiviral vectors as a vaccine platform against Infectious Diseases

The review gives a detailed summary of the groups work on LV vaccines against infectious diseases. As a review, it should cover also other groups work to this topic, which is lacking in the current manuscript.

In addition, a general understandable introduction into the topic should be included. The current introduction starts too detailed.

Vector production and potential safety problems coming from residual cell components should be discussed.

In addition, targeting of vectors might be a topic for discussion.

Round 2

Reviewer 2 Report

all issues solved.